# Enhancement of the Evaporation and Condensation Processes of a Solar Still with an Ultrasound Cotton Tent and a Thermoelectric Cooling Chamber

Naseer T. Alwan [1,2], Ayad S. Ahmed [1,3], Milia H. Majeed [1], Sergey E. Shcheklein [1], Salam J. Yaqoob [4,*], Anand Nayyar [5,6], Yunyoung Nam [7,*] and Mohamed Abouhawwash [8,9]

1 Department of Nuclear and Renewable Energy, Ural Federal University, 19 Mira St., 620002 Yekaterinburg, Russia; nassir.towfeek79@gmail.com (N.T.A.); ayadshmsallh@gmail.com (A.S.A.); milia.hameed88@gmail.com (M.H.M.); s.e.shcheklein@urfu.ru (S.E.S.)
2 Power Mechanics Department, Kirkuk Technical College, Northern Technical University, Kirkuk 36001, Iraq
3 Department of Refrigeration and Air Conditioning Engineering, Al-Rafidain University College, Baghdad 10001, Iraq
4 Department of Research and Education, Authority of the Popular Crowd, Baghdad 10001, Iraq
5 Graduate School, Duy Tan University, Da Nang 550000, Vietnam; anandnayyar@duytan.edu.vn
6 Faculty of Information Technology, Duy Tan University, Da Nang 550000, Vietnam
7 Department of Computer Science and Engineering, Soonchunhyang University, Asan 31538, Korea
8 Department of Mathematics, Faculty of Science, Mansoura University, Mansoura 35516, Egypt; abouhaww@msu.edu
9 Department of Computational Mathematics, Science, and Engineering, College of Engineering, Michigan State University, East Lansing, MI 48824, USA
* Correspondence: engsalamjabr@gmail.com (S.J.Y.); ynam@sch.ac.kr (Y.N.)

**Abstract:** In this paper, an experimental investigation study was conducted to show the effect of enhancing the evaporation and condensation processes inside a modified solar still by placing ultrasonic humidifiers inside a cotton mesh tent in the basin water and by installing a cooling chamber with thermoelectric elements on top of the solar still. Various parameters were recorded every hour, such as temperatures at different points within the solar still, the weather conditions (e.g., solar irradiance intensity, ambient air temperature, and wind speed), the yield of distilled water, and thermal efficiency on 29 July 2021 at the Ural Federal University (Russia). The production cost of distilled water from modified and traditional solar stills was also estimated. The experimental results showed that the productivity of the modified solar still increased by 124% compared with the traditional solar still, and the highest thermal efficiency was recorded at 2:00 p.m. (approximately 95.8% and 35.6% for modified and traditional solar stills, respectively). Finally, the productivity cost of distillate water (1 L) was approximately 0.040 and 0.042 \$/L for the modified and traditional solar stills, respectively. The current work has contributed to increasing solar still productivity by applying simple and new technologies with the lowest possible capital and operational costs.

**Keywords:** evaporation and condensation process; ultrasound cotton tent; thermoelectric cooling chamber; single modified solar still; distilled water; ultrasonic humidifiers

## 1. Introduction

Solar photovoltaic (PV) energy is a part of the family of clean, free, and renewable energy sources, and it can produce electrical and thermal energy. The PV system is a topology that is mostly used to generate electrical power globally with low mechanical installations compared with other renewable sources [1,2]. Although the PV system is a widely used source, it has several disadvantages, such as nonlinear characteristics; that is, it is dependent on solar irradiance and the ambient temperature [3–5]. PV panels can generate the required electrical power with only 10–15% global irradiance [6,7]. The increase in

the PV panel temperature can consequently lead to a decrease in the efficiency of the PV panel [8–10]. Therefore, several appropriate cooling mechanisms are used to remove the head partially in PV panels [11,12].

Water is essential for sustaining life. Although water covers 71% of the globe's surface, 97% of this is saltwater. Many countries suffer from a lack of potable water [13–15], as river water is not easily accessible. Several problems are also related to hygiene. Many countries (especially in the Middle East and Africa) suffer from a scarcity of potable water. Numerous attempts have been made to solve this deficit by extracting salt from seawater to obtain potable water.

In recent years, several researchers have used various techniques to solve drinking water shortages, such as reverse osmosis, vapor pressure, and electrolysis. The most recently applied methods are considered economical and do not have environmental side effects [16–19]. Solar PV energy is one of the easiest and most cost-effective sources for drinking water production and thermal applications such as water heating, cooling, and drying. Solar distillation systems for saline and untreated water are considered essential for reducing energy consumption [15]. However, the main disadvantage of solar water stills is low productivity. Therefore, several researchers have tried to use different methods and designs to increase productivity and thermal efficiency [16]. The modifications proposed are classified into passive and active methods. In passive solar stills, evaporation occurs without using additional external energy to heat the basin water and vice versa for active solar stills [17].

S. Nazari et al. [18] proposed enhancing solar still productivity by installing a single solar still on a parabolic dish and placing paraffin wax cells in the basin water. The results showed that the suggested improvement increased productivity during summer and winter by 65% and 45%, respectively. N.T. Alwan et al. [19,20] investigated the effect of changing the depth of the basin water from 1 cm to 3 cm on the performance and productivity of a single-slope solar still. The results showed that the heat transfer coefficients increased as the water depth decreased. The highest productivity was recorded at a depth of 1 cm, approximately 1.6 $L/m^2$, and 1.35 at 2 cm. The lowest productivity was recorded at 3 cm and approximately 1.05 $L/m^2 \cdot$day. A.S. Abdullah et al. [21] designed and constructed a modified piece of technology by installing a vertically rotating wick in a single-slope solar still and adding nanofluid to the basin water. This modification with and without nanofluid enhanced productivity by approximately 315% and 300%, respectively.

N.T. Alwan et al. [22] combined a solar still based on a single slope with a rotating hollow drum and integrated an external solar water collector into the modified system. This set-up enhanced the yield of distillate water by approximately 280% and 400% in summer and winter, respectively, compared with a conventional solar still. N.T. Alwan et al. [23] proposed a new combination to enhance evaporation and condensation inside a single-slope solar still. A diffusion absorption refrigerator was used so that the still ran day and night. The refrigerator's condenser was immersed in the basin water to increase its temperature, while the evaporator was placed on top of the solar still under a glass cover in a special box to enhance condensation. Compared with a traditional solar still, productivity improved by 251% and 469% during the day and night, respectively.

N.T. Alwan et al. [24] raised the productivity of a single-slope solar still by immersing three ultrasonic humidifiers in the water basin inside a cotton tent (wick) to improve evaporation. They demonstrated that the suggested modification improved the daily yield by 68% compared with a traditional solar still. N.T. Alwan et al. [25] proposed a practical validation based on a rectangular basin solar distillation. This study was based on a single slope using paraffin wax (PCM) cells. Moreover, a parabolic dish integrated with a central process unit-type solar water heater presented with a new biaxial tracking system was proposed [26]. In this system, the water temperatures at the outlet heater differed by about 20%, which means that the proposed system enhanced the temperature of the water for the central processing unit-type heat exchanger.

Sadeghi G. and Nazari S. [27] presented the purification of water using a hybrid nanomaterial and the application of the magnetic property to increase the rate of heat transfer. This study included two types of a solar still, one of which was conventional. The other was modified by using a cooling duct with four thermoelectric elements combined with an external evacuated tube-type solar collector. The results showed an improvement in the productivity of the modified solar still of about 218% and a 117% improvement in energy efficiency compared with a conventional solar still. S. Nazari et al. [28] conducted an experimental study to improve the evaporation efficiency of the solar distiller basin water using a copper oxide nanofluid ($Cu_2O$) and to increase the condensation efficiency by integrating the modified solar distiller with a cooling duct, on which four thermoelectric cooling units were installed. The results showed that the productivity and the energy efficiency of the modified solar still improved compared with a conventional solar distiller by approximately 81% and 112.5%, respectively. B. Mehdi. et al. [29] developed a predictive model for the efficiency of a solar still using a fuzzy neural heuristic system, improving the thermal conductivity of the aquarium water by adding nanoparticles ($Cu_2O$) and enhancing the condensation mechanism using a cooling duct with thermoelectric elements. The data were used as input for training AI methods. The results showed that the application of particle swarm optimization (PSO) remarkably improved the prediction performance over the rest of the proposed models.

It is therefore possible to state that the temperature differences between the evaporation and condensation surfaces inside the solar still have proven to be key in improving the productivity of the solar stills. It is for this reason that the studies reviewed above proposed different mechanisms to enhance the evaporation and condensation processes inside solar stills. It is, however, important to state that most of these mechanisms are bulky to construct and quite expensive, and hence new ways of enhancing the evaporation and condensation processes devoid of these negatives ought to be proposed. It is for that reason that the current study assesses the use of ultrasonic humidifiers and the thermoelectric cooling "Peltier cooler" technique to improve the yields of solar stills.

The main contribution of this paper is to improve the evaporation and condensation processes by using ultrasonic humidifiers in the basin water inside a cotton mesh cloth (wick) and to enhance condensation by installing a cooling chamber with thermoelectric coolers in Yekaterinburg, Russia. Moreover, high productivity at a relatively low cost is achieved. The cost of producing distilled water (1 L) with this modified solar still is analyzed in detail, and a comparison with previous studies in the same environmental conditions is provided.

The proposed paper is organized as follows. Section 2 presents the materials and methods. Section 3 analyzes the production cost. Section 4 highlights the experimentation and the performance analysis. Section 5 concludes the paper with the future scope.

## 2. Materials and Methods

This section describes the composition of the items used in building the test rig. The experiment took into consideration two models of solar stills: one known as the modified solar still module and the other known as the referenced module. Three ultrasonic humidifier elements are instilled inside the basin water solar still, and the cooling chamber module is integrated with the thermoelectric cooler elements, while the referenced module has no modifications.

### 2.1. Scheme of a Solar Distiller System

Figures 1 and 2 show a schematic diagram and a photographic view of the experimental installation. The traditional and modified solar distillers were made of 1.8 cm wooden MDF. Both distillers had the same dimensions (103.6 cm long, 53.6 cm wide, 61.8 cm on the long side, and 26.6 cm on the short side). The dimensions of the black galvanized water basin were as follows: 100 cm long, 50 cm wide, 10 cm deep, and 0.1 cm thick. The solar distillers were covered with plexiglass inclined at 35° degrees and with the following dimen-



sions: 103.6 cm long, 53.6 cm wide, 50 cm on the long side, 14.8 cm on the short side, and 0.3 cm thick. Aluminum channels were used to attach the plexiglass cover onto the body of the solar still. After the water vapor condensed on the inner surface of the plexiglass cover and the aluminum plate of the cooling chamber, the distilled water flowed through the aluminum channels to the graduated vessel beneath the solar still. Silicone glue was used to fix all the parts together [17]. A polycrystalline photoelectric panel of 18.60 V DC and 5.92 A and a peak power of 110 W was selected, with dimensions L = 101.5 cm, W = 66.8 cm, and H = 3 cm. The PV panel was tilted at an angle of 35° by a mechanism consisting of a base of four mild steel irons moving at different angles (supporting structure).

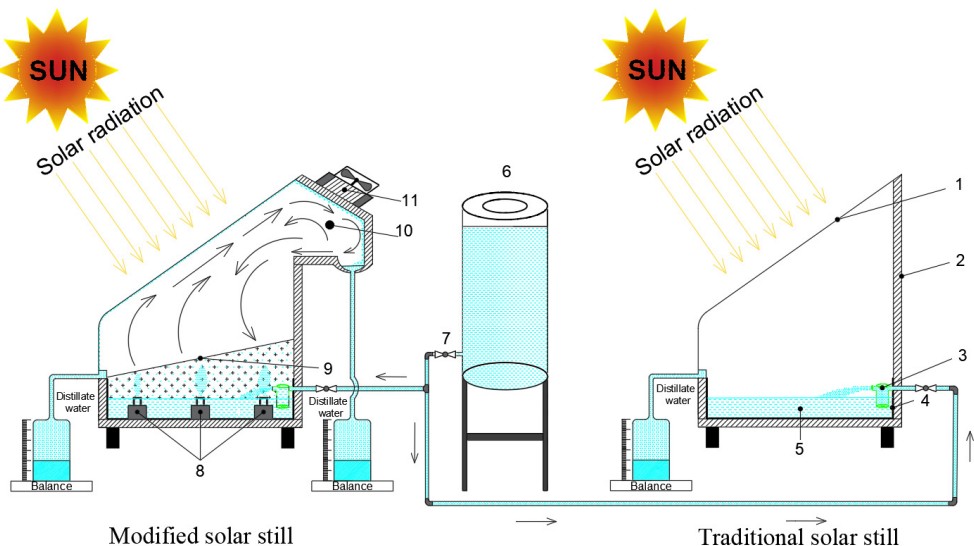

**Figure 1.** Schematic diagram of modified and traditional solar stills: 1 = plexiglass cover, 2 = MDF wooden panel, 3 = mechanical floater, 4 = basin, 5 = water, 6 = water tank, 7 = globe valve, 8 = ultrasonic humidifier elements, 9 = cotton tent, 10 = cooling chamber, and 11 = thermoelectric cooling elements.

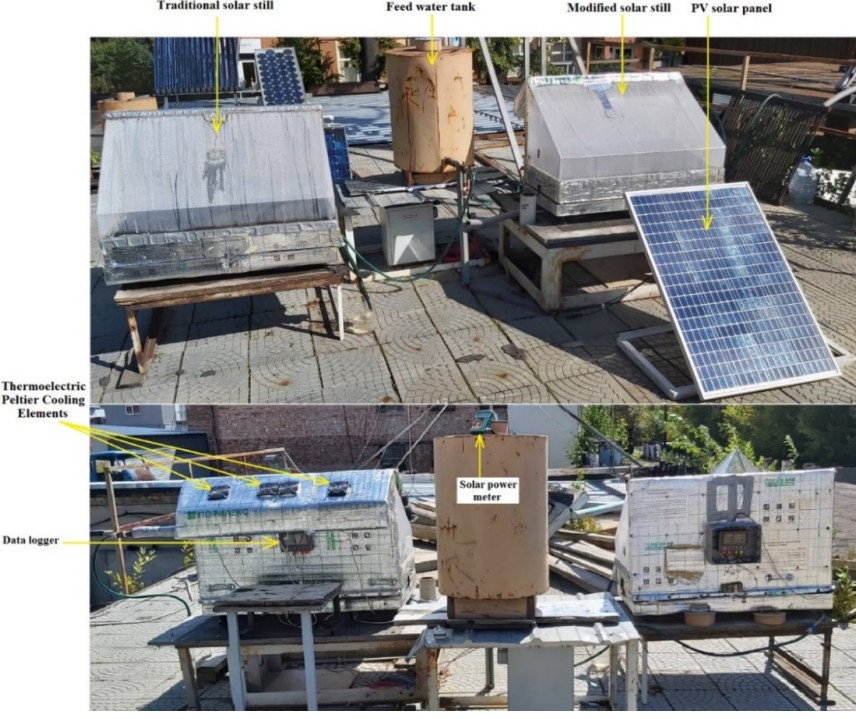

**Figure 2.** Photographic view of modified and traditional solar stills.

### 2.1.1. Working Principle for Ultrasonic Humidifiers

Ultrasonic humidifiers use a metal diaphragm to convert a high-frequency electronic signal into a high-frequency mechanical oscillation that disperses the water layers into mist droplets. To increase the interfacial surface and the basin water evaporation efficiency inside the modified solar still, three ultrasonic humidifiers were placed in the basin water inside a polycarbonate frame covered by a black cotton mesh (wick). The proposed hybrid technology led to reducing the distance between the evaporation and condensation surfaces (water and glass cover), thus increasing the rate of heat and mass transfer within the modified solar still as well as reducing the depth of the basin water to a filmy layer which quickly evaporated from the surface of the cotton tent, which was constantly renewed by ultrasonic humidifiers compared with the depth of the basin water in a conventional solar still, as shown in Figure 3.

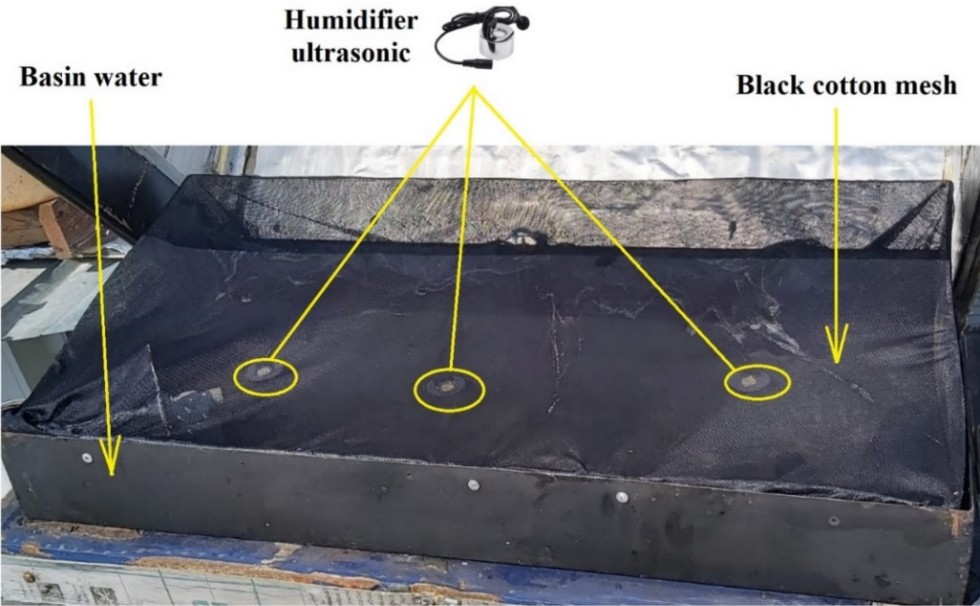

**Figure 3.** Photographic view of water basin modified with ultrasonic humidifiers inside the cotton mesh.

### 2.1.2. Working Principle for Thermoelectric Coolers

The direct conversion of temperature differences into an electric voltage and, conversely, converting the electric voltage into a temperature difference is known as the thermoelectric effect. A thermoelectric cooler (TEC), which is also known as a "Peltier cooler", employs the Peltier effect for heat exchange. A TEC is made up of P- and N-type semi-conductor couples [30,31]. The cold part of the TEC is fixed at the rear side of the aluminum plate, while the heat sink is instilled on the hot part of the TEC. The heat generated from the TEC by the forced convection is transferred to the ambient air with the help of a 12-V DC fan. In this case, the aluminum plate (cooling chamber) would be cooled by the TEC as depicted in Figure 4. The working principle of the TEC is presented in Figure 5.

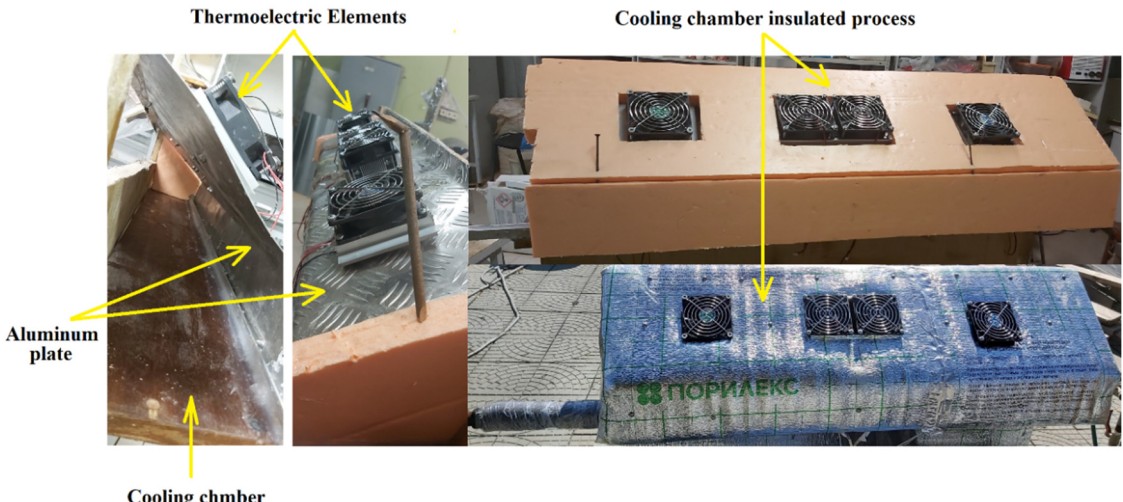

**Figure 4.** Photographic view of the cooling chamber.

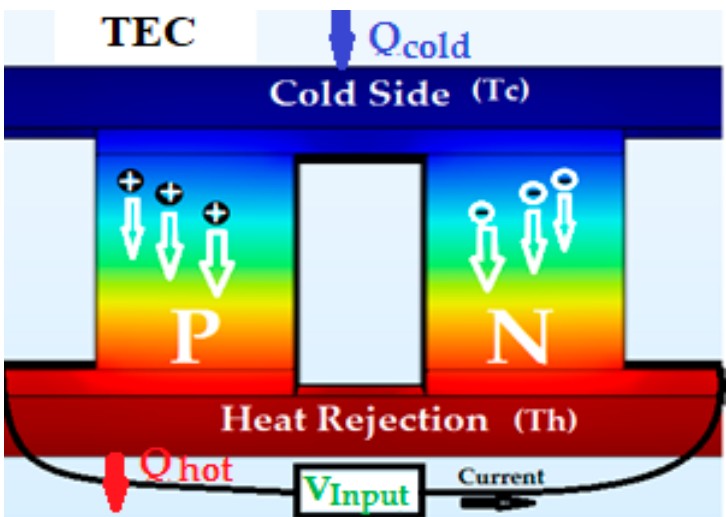

**Figure 5.** Working principle of the TEC [30].

The thermoelectric module (TEM) uses the Peltier effect to provide a cooling or heating effect. The Peltier cooler elements enable heat transfer from one part to another part of the Peltier module based on the direction of the current. The principle behind its work is to generate a heat flux between the P-N junction, depending on the voltage difference between the two of these parts. Therefore, a voltage is applied across the device to make one of the sides hot while the other becomes cold (side A and side B). The rate of heat generated $\dot{Q}$ can be expressed as indicated in Equation (1) [32]. The properties and parameters of the TEC are presented in Table 1. A cooling chamber made of an aluminum plate (0.1 cm thick) was installed on top of the solar still. Four thermoelectric cooling elements (Peltier elements) were installed to improve condensation, as shown in Figure 4:

$$\dot{Q} = (\Pi_A - \Pi_B) \times \mathrm{I} \tag{1}$$

where $\Pi_A$ and $\Pi_B$ are the Peltier coefficients of conductors A and B, respectively, and I is the electric current from A to B.

**Table 1.** Description of the thermoelectric cooler (TEC) and DC fan.

| Description | Feature |
|---|---|
| Model | AV-F9025MS |
| Name | DC12V BRUSHLESS FAN |
| Rated Voltage | 12 V DC |
| Operation Voltage | DC 6.8–12.8 V |
| Consuming Current | 0.20 A |
| Consuming Power | 1.2 W |
| Rated Speed | 3000 R.P.M. |
| Max. Air Flow | 32.28 CFM |
| Size | $80 \times 80 \times 25$ mm |
| Weight | 40 g |

### 2.2. Experimental Procedure

The study was conducted at the Ural Federal University in Yekaterinburg, Russia (56.8° N and 60.6° E) in July 2021 from 8:00 a.m. to 8:00 p.m. At the start of each test, the depth of the basin water was kept constant at 5 cm, which was suitable for the immersion of the ultrasonic humidifiers. A k-type thermocouple with an 88598-type four-socket data logger was used to measure the temperatures of the basin liner, basin water, surface of the cotton mesh, plexiglass cover, and inner surface of the cooling chamber. A TM-207 radiation intensity meter was used to measure the intensity of the solar radiation, and a UT 363 BT anemometer was used to measure the wind speed.

To measure the hourly production of distillate water from the cooling chamber and the plexiglass cover, two graduated cylinders with a capacity of 1 L were used. To study the properties of the water before and after solar distillation, E-1 TDS and EC meters were used to measure the total dissolved solids and electrical conductivity. A PH-05 pH meter was used to measure the water's hydrogen potential. Four thermoelectric cooling elements (Peltier elements) were used to cool the external condensing chamber. The thermoelectric cold side was installed on the aluminum plate (condenser), and the thermoelectric hot side was installed on the finned heat sink and cooling fan with special thermal silicon. The rest of the cooling chamber was tightly sealed.

### 2.3. Uncertainty Analysis

The accuracy of each device used in the experimental measurement was calculated to determine the uncertainty values accurately [25]. Table 2 includes data on the accuracy of each device used as well as the error range, according to the equations mentioned in previous studies [26,33,34].

**Table 2.** Results of experimental device uncertainty analysis.

| Device | Accuracy Value | Measuring Range | Error Ratio % | Measuring Units |
|---|---|---|---|---|
| Temperature data recorder | 1 °C | from −200 to 1370 | 0.3% | °C |
| K-type thermocouple | 0.1 °C | from −100 to 200 | 0.3% | °C |
| TM-207 radiation intensity meter | 0.1% | 0–2000 | 0.1% | $W/m^2$ |
| UT 363 BT anemometer | 0.05 | 0–30 | 3% | m/s |

*2.4. Thermal Efficiency*

The thermal efficiency of the modified solar still was calculated based on the following equation:

$$\eta_{th} = \frac{\dot{m}_{ev} * A_{bp} * h_{fg}}{[I(t) * A_{bp} + n_{PE} * P_{PE} + n_{fan} * P_{fan} + n_{UM} + P_{UM}] * 3600} 100\% \qquad (2)$$

$$h_{fg} = 100 \times [2501.9 - 2.407 \times T_w + \frac{1.1922 \times T_w^2}{10^3} - \frac{1.586 \times T_w^3}{10^5}] \qquad (3)$$

where $\eta_{th}$ is the thermal efficiency, $\dot{m}_{ev}$ is the hourly yield of distillate water (L/m$^2$/h), $h_{fg}$ is the latent heat of the basin water at an average temperature (J/Kg), $A_{bp}$ is the surface area of the water basin (m$^2$), $I(t)$ is the intensity of the solar radiation (W/m$^2$), $n_{PE}$, $n_{fan}$, and $n_{UM}$ are the number of Peltier elements, fans, and ultrasonic humidifiers, respectively, and $P_{PE}$, $P_{fan}$, and $P_{UM}$ are the power consumption of the Peltier elements, fans, and ultrasonic humidifiers, respectively.

**3. Production Cost Analysis**

In water desalination systems, several goals must be met during design and implementation. The most important one is the quality of the structural materials and the production cost per liter of distilled water (PCD) in \$/L·m$^2$, which is calculated as follows [22]:

$$PCD = \frac{TCY}{\dot{m}_{ev(year)}} \qquad (4)$$

$$\dot{m}_{ev(year)} = \sum_{i=1}^{i=180} \dot{m}_{ev(day)} \qquad (5)$$

where TCY is the total cost per year (\$/L·year), $\dot{m}_{ev(year)}$ is the average freshwater productivity per year (L/m$^2$·year), and $\dot{m}_{ev(day)}$ is the average freshwater productivity per day (L/m$^2$·day). This work assumes that Yekaterinburg has 180 sunny days [22]. Thus, the total cost per year is calculated as follows [22]:

$$TCY = FCY + MCY - SVY \qquad (6)$$

$$FCY = FCR \times C \qquad (7)$$

$$FCR = i(i+1)^n \times [(i+1)^{n-1}]^{-1} \qquad (8)$$

$$MCY = 15\% \times FCY \qquad (9)$$

$$SVY = FSF \times SV \qquad (10)$$

$$FSF = i \times [(i+1)^{n-1}]^{-1} \qquad (11)$$

$$SV = 20\% \times C \qquad (12)$$

where FCY is the fixed cost per year, MCY is the maintenance cost per year, SVY is the salvage value of the solar distiller per year, FCR is the factor of capital recovery, C is the capital cost of the solar distiller, i is the interest rate per year (12%), n is the average life expectancy of the solar still (10 years), FSF is the factor of a sinking fund, and SV is the salvage value.

The estimated capital cost of the components is shown in Table 3. The estimates amounted to \$82 and \$178 for traditional and modified solar stills, respectively. Table 4 includes a detailed distillate water productivity cost analysis (1 L) from conventional and modified solar stills, which amounted to \$0.042 and \$0.040, respectively. The current work suggests that the solar distiller model operates 180 days per year (i.e., the average number

of sunny days per year in Yekaterinburg, Russia), whereas other studies (other people's works) suggested 365 days a year with various weather conditions. Compared with previous studies, the cost of producing modified solar distillers in this work is comparable (acceptable) to that recorded in previous studies, as shown in Table 5.

**Table 3.** The capital cost of solar still components.

| Components | Traditional Solar Still | Modified Solar Still |
|---|---|---|
| Wooden board | 14 | 14 |
| Plexiglass cover | 15 | 15 |
| Galvanized stainless steel | 11 | 11 |
| Solar panel | - | 60 |
| Peltier element | - | 20 |
| Ultrasonic elements | - | 15 |
| Cotton mesh | - | 1 |
| Various materials and accessories | 42 | 42 |
| Total cost ($) | 82 | 178 |

**Table 4.** Distillate water productivity cost analysis ($).

| Expression | Traditional Solar Still | Modified Solar Still |
|---|---|---|
| C ($) | 82 | 178 |
| FCR | 0.1769 | 0.1769 |
| FSF | 0.0569 | 0.0569 |
| FCY | 14.505 | 31.48 |
| MCY | 2.175 | 4.72 |
| SV | 16.4 | 35.6 |
| SVY | 0.93 | 2.02 |
| TCY | 15.747 | 34.17 |
| $\dot{m}_{ev(day)}$ | 2.095 | 4.7 |
| $\dot{m}_{ev(year)}$ | 377.1 | 846 |
| PCD | 0.042 | 0.040 |

**Table 5.** Comparison of production costs with previous studies in the same environmental conditions.

| Study | Type of Single-Slope Solar Still Enhancement | Study Location | Daily Yield of Distillate Water (L/m²·day) | Cost of Productivity ($/L) |
|---|---|---|---|---|
| [17] | Solar still with external solar collector | Russia | 5.5 | 0.047 |
| [19] | Solar still at different water depths | Russia | 1.6 at 1 cm | 0.033 |
| [22] | Solar still integrated with hollow drum | Russia | 12.5 | 0.026 |
| [23] | Solar still combined with diffusion absorption cooling | Russia | 5.180 | 0.046 |
| [24] | Solar still combined with ultrasonic humidifiers | Russia | 4.2 | 0.0259 |
| [35] | Single-slope solar distiller | Pakistan | 3.2 | 0.062 |
| [36] | Single-slope solar still combined with ultrasonic humidifier and integrated with solar water heater | Egypt | 7.4 | 0.037 |
| [37] | Single-slope solar still integrated with rotating drum | Saudi Arabia | 11 | 0.039 |
| [38] | Hybrid (PV/T) active single-slope solar stills | India | 1.90 | 0.14 |

**Table 5.** *Cont.*

| Study | Type of Single-Slope Solar Still Enhancement | Study Location | Daily Yield of Distillate Water (L/m$^2$·day) | Cost of Productivity ($/L) |
|---|---|---|---|---|
| [25] | Single-slope solar distiller combined with thermal storage materials (PCM) | Iraq | 2.35 | 0.035 |
| [39] | Single-basin single-slope with a solar collector | Jordan | 4.78 | 0.115 |
| [40] | Single-basin single-slope with a separate condenser | Turkey | 6 | 0.06 |
| Current study | Solar still combined with ultrasonic cotton mesh and Peltier cooling chamber | Russia | 4.7 | 0.040 |

## 4. Experimentation and Performance Analysis

This section describes the effect of the two technologies that have been used to improve the productivity of a solar still. The interface and the evaporation efficiency inside the solar still were increased by using ultrasonic humidifiers in the basin water inside a mesh tent (wick). The temperature in the condensing area in the upper part of the solar still was reduced by installing an aluminum channel cooled by the thermoelectric elements. Thus, the effect of environmental parameters such as the solar radiation, ambient temperature, and wind speed along with the effect of the design parameters (e.g., ultrasonic humidifier cotton tent and thermoelectric cooling chamber) on the performance and the productivity of solar stills were discussed. The performance of the solar distillers on rainy or cloudy days was not considered, because the highest productivity was recorded for a sunny day (typical day), and solar distiller productivity depends on the intensity of solar radiation and the ambient air temperature [34]. Therefore, the tests were conducted hourly from 8:00 a.m. to 8:00 p.m. (12 tests) on 29 July 2021, a typical sunny day.

Figure 6 shows the weather conditions (intensity of solar radiation, ambient air temperature, and wind speed) for the test day. The solar radiation I(t) and ambient temperature (Ta) were relatively low in the early hours and then gradually increased until the period from 12:00 p.m. to 2:00 p.m., reaching approximately 923 W/m$^2$. The solar radiation and ambient temperature then decreased until sunset. Solar radiation reaching the Earth's surface needs time to transfer its heat energy to nearby surfaces, such that the maximum ambient air temperature was achieved at around 4:00 p.m., reaching approximately 34.3 °C. The wind speed (Va) was uneven, and the highest value was recorded in the afternoon.

Figure 7 shows the hourly temperature differences of the basin liner ($T_{bp}$), basin water ($T_{bw}$), plexiglass cover ($T_g$), mesh cloth ($T_{cloth}$), inner surface of the cooling chamber ($T_{cooler}$), and intensity of solar radiation I(t) of the modified and traditional solar stills for 29 July 2021. The temperature of the plexiglass cover and the basin water in the traditional solar still were close during the morning hours. Over time, the difference between the temperatures increased, as the heat capacity of water is higher than that of plexiglass. The highest basin water temperature was found at 2:00 p.m., the time of peak solar radiation intensity. For the modified solar still, the temperature of the mesh cloth (wick), which was moistened by a water film (aerosol) providing flowing water from the ultrasonic humidifiers, was always higher than that of the basin water in the traditional solar still due to the thinness of the wick's water layer; that is, this needed a small amount of time to raise its temperature compared with the depth of basin water (5 cm) in the traditional solar still. This figure also shows that the temperature of the inner surface of the cooling chamber was lower than that of the plexiglass in the modified and traditional solar stills because of the thermoelectric elements, which decreased the temperature of the cooling chamber's condensing surface. Therefore, the low temperature of the cooled aluminum plate of the cooling chamber, which decreased by about 7–11 °C compared with the plexiglass cover, indicates the positive effect of the proposed cooling mechanism, as shown in Figure 7.

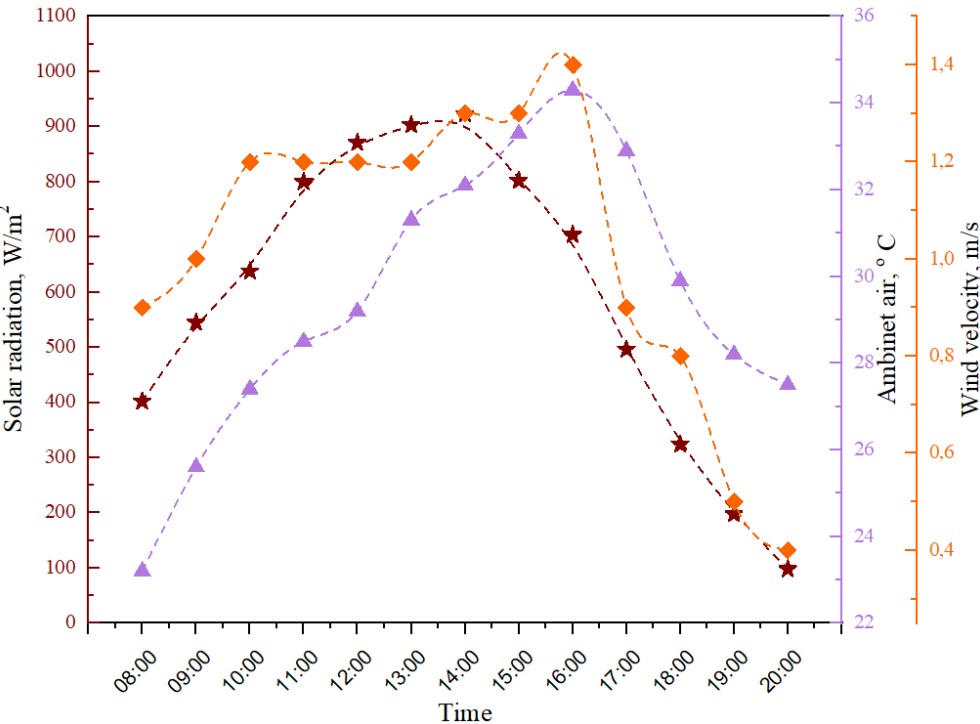

**Figure 6.** Change per hour in solar radiation intensity, ambient air temperature, and wind speed for 29 July 2021.

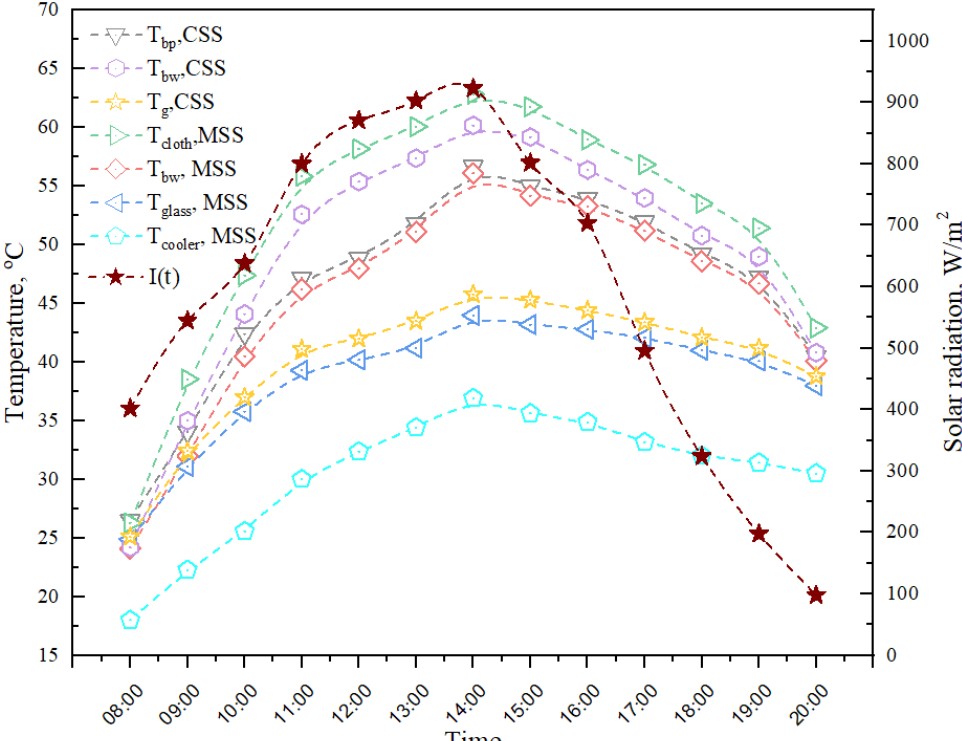

**Figure 7.** Hourly change of temperatures at different points in modified and traditional solar stills for 29 July 2021.

In solar distillers, two conditions must be achieved to increase productivity, namely a high evaporation rate for the basin water and a high condensation rate for the generated water vapor, thus increasing the temperature difference between the evaporating and condensing surfaces (increasing the rate of heat and mass transfer). An increase in the

rate of heat and mass transfer is a result of an improvement in the convective heat transfer coefficient ($h_c$) and evaporative heat transfer coefficient ($h_{ev}$) [20].

Figure 8 represents the hourly change in distillate water output from the modified and traditional solar stills and the intensity of the solar radiation. The distillate water yield was directly proportional to the intensity of the solar radiation during the day. Thus, the behavior of the yield curves for both stills was approximately the same as that of the solar radiation intensity. Therefore, the highest rate of productivity per hour was recorded at 2:00 p.m. at the highest value of solar radiation, being approximately 775 mL/m$^2$/h for the modified solar still and 420 mL/m$^2$/h for the conventional solar still. These results show an improvement of 84% because of the ultrasonic humidifiers continuously moistening the cotton wick, thus increasing the rate of basin water evaporation. The cooling chamber also played a role in accelerating the condensation of water vapor on its surface. Therefore, productivity was enhanced by increasing the temperature difference between the evaporation and condensation surfaces (increasing the value of the evaporative heat transfer coefficient by increasing the heat transfer rate via natural convection).

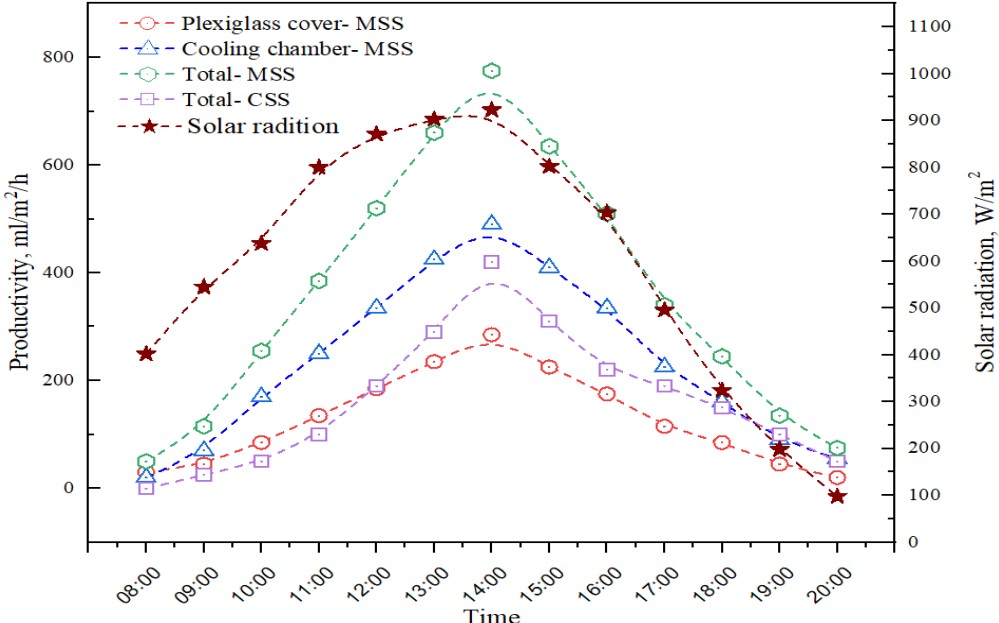

**Figure 8.** Hourly change in the productivity of modified and traditional solar stills for 29 July 2021.

Figure 9 shows that the cumulative production of distillate water from 8:00 a.m. to 8:00 p.m. for the modified solar still was more than that of the traditional solar still by approximately 124% (4.7 and 2.095 L/m$^2$·day from the modified and traditional solar stills, respectively) due to the high condensation rate (effect of ultrasonic humidifiers inside the mesh tent) and the ability of the cooling chamber to condense more water vapor on its inner walls.

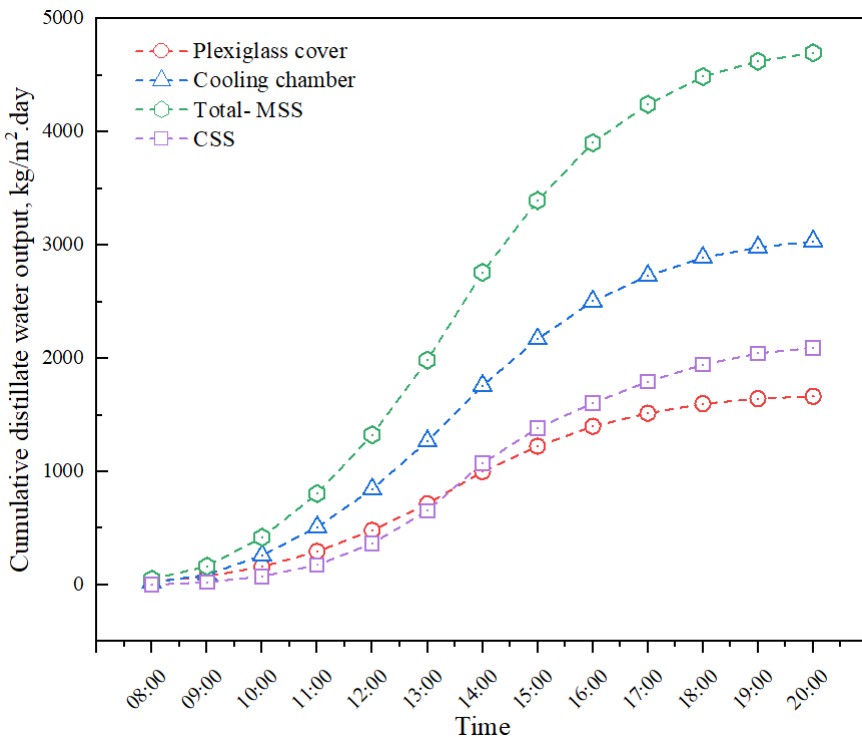

**Figure 9.** Cumulative daily distillate water from modified and traditional solar stills for 29 July 2021.

Figure 10 shows the hourly change in the thermal efficiency of both solar stills. The thermal efficiency of the modified solar still was always greater than that of the traditional solar still, as shown in Equation (1) [19].

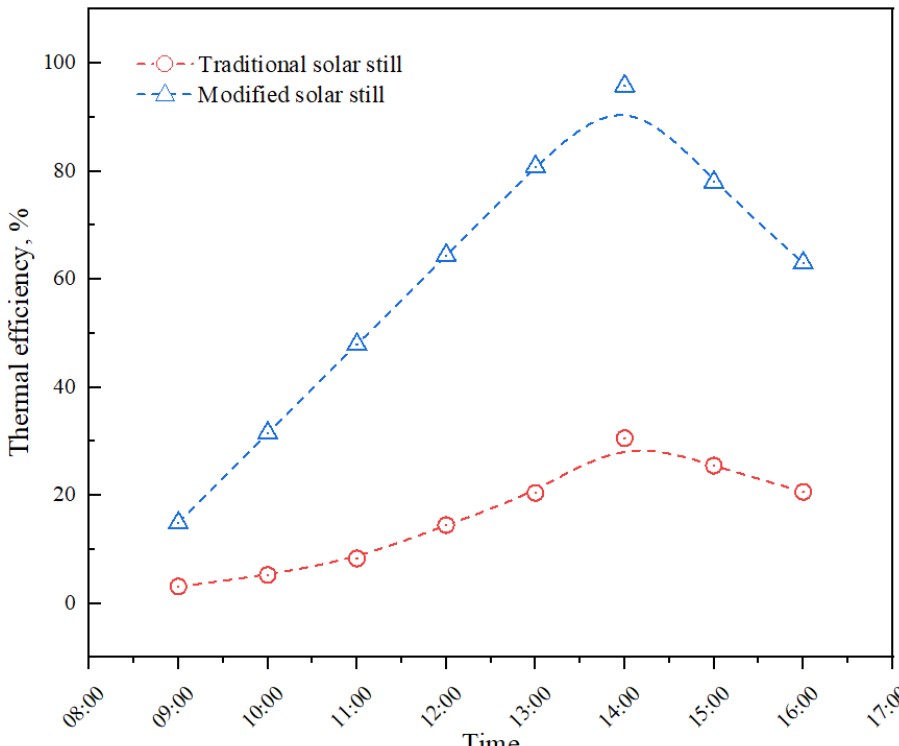

**Figure 10.** Hourly thermal efficiency of modified and traditional solar stills for 29 July 2021.

Additionally, the present experimental work included an examination of the distilled water samples, including the value of the total dissolved solids (TDS), which ranged between 13 and 26 ppm.

## 5. Conclusions

In this paper, an experimental investigation was conducted to show the effect of enhancing the evaporation and condensation processes inside a single modified solar still by placing ultrasonic humidifiers inside a cotton mesh tent in the basin water and by installing a cooling chamber with thermoelectric elements on top of the still. The result's highlights can be summarized as follows:

(1) The production of distilled water in the cooling chamber was enhanced by installing thermoelectric Peltier cooling elements on its walls. This set-up increased the condensation of water vapor in the channel of the cooling chamber. Productivity was also improved by placing ultrasonic humidifiers inside a cotton mesh tent in the basin water. The productivity of the modified solar still increased by 124% compared with the traditional solar still.

(2) The thermal efficiency of the modified solar still was always higher than that of the traditional solar still over 12 h due to the increase in productivity, which itself was a result of improving the evaporation and condensation processes.

(3) The low temperature of the cooled aluminum plate of the cooling chamber, which decreased by about 7–11 °C, compared with the plexiglass cover indicates the positive effect of the proposed cooling mechanism.

(4) The productivity cost of distillate water (1 L) was approximately 0.040 and 0.042 \$/L for the modified and traditional solar stills, respectively. The economic analysis shows that the proposed modification agreed with previous studies in the environmental conditions of Yekaterinburg, Russia in terms of the cost of producing distilled water.

(5) Therefore, it can be concluded that the ultrasonic humidifiers and thermoelectric cooling elements were effective, considering the parameters assessed, and could be used to enhance the productivity of the solar stills in hot climatic areas where water is scarce.

(6) The results showed that the temperature of the basin water in the modified solar still was lower than that of the conventional solar still. Therefore, to overcome this problem, it is recommended in the future to combine the modified still with an external solar collector to increase the temperature of the basin water under the cotton tent.

**Author Contributions:** Conceptualization, N.T.A., A.S.A. and S.E.S.; methodology, N.T.A., A.S.A. and M.H.M.; software, N.T.A., A.S.A. and M.H.M.; validation and formal analysis, A.S.A., A.N., N.T.A. and M.A.; investigation, S.J.Y., A.N., Y.N. and M.A.; resources, N.T.A., S.E.S. and M.A.; data curation, N.T.A., S.E.S., A.S.A. and S.J.Y.; writing—original draft preparation, N.T.A., A.S.A., M.H.M. and S.E.S.; writing—review and editing, N.T.A., S.J.Y., A.N. and Y.N.; visualization, N.T.A., S.J.Y., A.N. and M.A.; supervision, S.E.S., A.S.A. and M.H.M.; project administration, S.E.S.; funding acquisition, Y.N. All authors have read and agreed to the published version of the manuscript.

**Funding:** This research was supported by a grant of the Korea Health Technology R&D Project through the Korea Health Industry Development Institute (KHIDI), funded by the Ministry of Health & Welfare, Republic of Korea (grant number: HI21C1831) and the Soonchunhyang University Research Fund.

**Institutional Review Board Statement:** The study did not involve humans or animals.

**Informed Consent Statement:** The study did not involve humans.

**Conflicts of Interest:** The authors declare no conflict of interest.

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
