# Peer review of "Enhancement of the Evaporation and Condensation Processes of a Solar Still with an Ultrasound Cotton Tent and a Thermoelectric Cooling Chamber"

_electronics, doi:10.3390/electronics11020284_

Round 1

Reviewer 1 Report

1- Expand the conclusion section further and quantify the results of this section. The number of cases (conclusions) should not be less than 5. 

Author Response

Dear respected editor,

Regarding the revision for manuscript number 1528730 with the title “Enhancement of the Evaporation and Condensation Processes of a Solar Still with an Ultrasound Cotton Tent and a Thermoelectric Cooling Chamber’’

I would like to thank you for your useful feedback on my manuscript and appreciate the constructive comments of the reviewers. I have carefully read through the reviewers’ comments. All suggested comments by the reviewers were added to the manuscript file. All suggested comments were considered point by point.

I am looking for the acceptance of the manuscript soon.

Best regards,

Salam J. Yaqoob

Reviewer 2 Report

Enhancement of the Evaporation and Condensation Processes of a Solar Still with an Ultrasound Cotton Tent and a Thermoelectric Cooling Chamber

Under the present manuscript, the effect of ultrasonic humidifier inside the cotton mesh tent and cooling chamber with thermoelectric elements on top of the solar still to enhance evaporation and condensation process is experimentally investigated. As the fresh water scarcity is increasing and desalination systems can play a substantial role to overcome the issue, the present study may be useful to improve the performance of solar still. However, there are several points that should implemented before final acceptance of this manuscript and therefore, I am recommending this manuscript for major revisions.

The comments/suggestions that should be addressed are as follow:

  1. It is suggested to include a statement regarding how the results presented in the manuscript may be useful for the development of solar stills in the end of abstract.
  2. The authors should modify the last paragraph of the introduction section and research gaps based on the literature review, objective of the present study and outline of the present manuscript should be included.
  3. It is suggested to include a subsection of uncertainty analysis in section 2 as the results obtained through experimental approach are insignificant without it.
  4. What is the utility of wind velocity measured using anemometer for the calculation of performance parameters in the present study.
  5. Is the present experimental study also carried out for water samples with different values of total dissolved solids (TDS)? What range of TDS in water can be treated through the investigated modified solar still?
  6. Why authors have presented the results in terms of thermo-hydraulic efficiency in spite of thermal efficiency as the pressure drop has significant impact over the performance thermal systems?
  7. Why the magnitude of cumulative distillate water output is changing with different rate in plexiglass cover of modified solar still than conventional solar still as shown in Fig. 7?

It is suggested to include a paragraph in the end of the conclusion section about how the present study may be useful in future?

Author Response

(The authors gave the same response as above.)

Reviewer 3 Report

  1. In the article, the main attention is given to the costs of obtaining water using solar panels. But that is not his main destiny. Traditionally, however, solar is used to produce energy. The authors plan to combine these two functions in some way. It would be worth presenting not only the costs of obtaining distilled water and the related benefits. However, it is more clearly stated that the amount of energy obtained from solar will not decrease. How is this guaranteed? And if the amount of energy can go down, does the overall balance of such collaborative action make sense? Maybe it is worth using a separate, specialized device for the production of water?
  2. The Materials and Methods section presents the research scheme and procedure. Virtually nothing has been written about the methods that are used. It would be worth presenting them more precisely
  3. In the list of authors (line 5-6), only one author is marked for correspondence (*), but below, on line 19, there are two corresponding authors. You have to put a symbol (*) for 2 people
  4. When citing, some authors are given with initials and others without (for example lines 64 and 76). It has to be done in the same way. The remark applies to all text
  5. If there is a decision to quote surnames with initials, they must be given everywhere or before or after the surname. Now in the article, once there, once there. Example – lines 86 and 96. The remark applies to all text
  6. In some places there is a dot after the surnames (inside a sentence). Not needed. For example – line 72 - Abdullah. et al. [21] designed …
  7. In some places there is a comma after the surnames (inside a sentence). Not needed. For example – line 96 - Sadeghi, and Nazari, S. …..
  8. In some sentences the colon (:) is unnecessary followed by a capitalized word. It doesn't make sense. For example – line 98 - This study included two types of solar still: One was conventional. The remark applies to all text
  9. In Cu2O entries, it is better to use the subscript and write as Cu2O (lines 104, 110 and others)
  10. Oversized fonts and other bad styles are used in all tables. Please adapt to the journal format
  11. Why are some units in italics but others not? Please do one way. Example - the last two rows in table 1 - ?/?2 and  m/s , or lines 194-199. The remark applies to all text
  12. The same size of letters must be used in all formulas - otherwise it is very difficult to understand where the main letters are and where the indexes are. For example – lines 192, 193, 204, 205 and others. The remark applies to all text
  13. Why are some formulas italicized and others not? The same question applies to all letters for variable and fixed values. For example – lines 191-220. The remark applies to all text
  14. In the dimension units (L/m2.day or L/m2.hr), the dot should be given not at the bottom but in the middle of the row - L / m2 · day or  or L/m2 · hr
  15. One and the same unit in different places, written in different ways - L/m2.day (line ) and L/m2/day (line 233 - table header). The remark applies to all text
  16. To be compliant with SI, the symbol used for hour must be h, not hr. It will be good variant write not L/m2.hr, only L/m2/h
  17. The formatting of references does not meet the requirements of the journal. See Template.
  18. You need to add a DOI to any publications which have it
  19. If publication doesn't have a DOI, it's a good idea to add an access link

Author Response

(The authors gave the same response as above.)

Round 2

Reviewer 2 Report

The aforementioned manuscript has now been reviewed and the authors have addressed all the comments satisfactorily. Therefore, this manuscript can now be recommended for publication.

Reviewer 3 Report

Accept in present form

This manuscript is a resubmission of an earlier submission. The following is a list of the peer review reports and author responses from that submission.

Round 1

Reviewer 1 Report

The authors have implemented an interesting work on techniques for enhancement of evaporation and condensation rates in solar stills. The state-of-the-art is convincing enough for publication; however, some comments are stated in order to enhance the overall quality of the manuscript before publication:

1- The writing and the language of the paper needs to be modified.

2- Do a check and add nomenclature if it need.

3- In each part, a good classification is done. The literature review and the bibliography of this work must be improved by adding more references about mechanisms of evaporations and condensations in solar stills which is related to manuscript. Moreover, some suggestions are as follows:

  • Retrofitting a thermoelectric-based solar still integrated with an evacuated tube collector utilizing an antibacterial-magnetic hybrid nanofluid, doi: 1016/j.desal.2020.114871.
  • Performance improvement of a single slope solar still by employing thermoelectric cooling channel and copper oxide nanofluid: an experimental study, https://doi.org/10.1016/j.jclepro.2018.10.194.
  • Modeling of energy efficiency for a solar still fitted with thermoelectric modules by ANFIS and PSO-enhanced neural network: A nanofluid application, https://doi.org/10.1016/j.powtec.2021.03.001.

4- Figure 1 shows a photovoltaic panel that is not mentioned anywhere in the text. If the PV panel is used in experimental tests, be sure to explain it in the text.

5- In Figure 2 (solar still schematic), a number is written to describe the parts on the figure, but the explanation of this number is not seen in the text or table!!!

6- Describe the effect of the ultrasonic humidifiers on the distillation mechanism in the text. In order to know how the ultrasonic humidifiers affects the solar still basin? also explain how ultrasonic humidifiers works?

7- You have stated the equations used in the solar still in the results section. Express the equations separately and separate them from the results section.

8- In the equations section (line 162) you have stated that we have sunny days in 180 in Yekaterinburg. Be sure to include a reference for this sentence.

9- In Figure 6, there is no legend for solar radiation. Insert the radiation legend in the figure.

10- Why did you put the thermoelectric arrangement on the condensing plate like this? I mean, two of the thermoelectrics are stuck in the middle and two are on the sides. Spread the thermoelectrics evenly over the surface so that the temperature cools evenly.

11- In the last paragraph of the introduction, describe the innovation of the modified solar still in more details.

12- Expand the conclusion section further and quantify the results of this section.

13- If you do not have a time limit and it is important for you to improve the quality of the article, also state the exergy analysis.

Author Response

Dear Reviewer

I would like to thank you for your useful feedback to my manuscript and appreciate the constructive comments of the reviewers. I have carefully read through the reviewer comments, and provided accurate constructive responses.

Reviewer 2 Report

In this manuscript, the authors reported on a modified solar still that included ultrasonic humidifiers and cooling components, claiming that the water yield had been increased. During the outdoor testing, several experimental data was acquired, including stream temperature and irradiation intensity, and the water productivity cost was evaluated. This work appeared to merely show some experimental results with few explanations. And the referee suggests the paper should be further modified to meet the Journals' requirements. Following are the concerns:

  1. Using an ultrasonic humidifier to speed up the formation of water streams and utilizing cooling devices to lower the temperature at the condensation interface are well-known methods for stream generation enhancement. So, what is the key point of the present work?
  2. The entire text was merely a description of some experimental findings; the findings should be thoroughly discussed with appropriate references.
  3. Why were there four humidifiers in different positions in the system? Is this optimized in any way? or any modeling results that account for this? To make your design work, you should do more control experiments.
  4. What about the presentation of other people's work? The authors are unable to only discuss their own findings. More comparison should be added, so that others can discuss your system.
  5. If the entire study was based on a single day's results, how about the stability in this system? What about performance on rainy or overcast days, for example?
  6. The manuscript contains some grammatical errors, and the format of the manuscript should be modified to make it more accessible to readers.

Author Response

(The authors gave the same response as above.)

Round 2

Reviewer 1 Report

Necessary corrections were made.

Thanks

Reviewer 2 Report

In the revised manuscript, the authors added some remarks regarding the experimental results, while the referee's core concerns, such as the design principle, long-term stability and environmental influences, were not articulated. Also, no additional comparisons with previous research results have been discussed. The novelty was not high, the controlling trials were missed, and the discussion was insufficient. Therefore, the referee suggested rejection of the work at present stage.